# Neonatal Subcutaneous BCG Vaccination Decreases Atherosclerotic Plaque Number and Plaque Macrophage Content in *ApoE*^−/−^ Mice

**DOI:** 10.3390/biology11101511

**Published:** 2022-10-15

**Authors:** Siroon Bekkering, Krishan Singh, Hui Lu, Albert P. Limawan, Claudia A. Nold-Petry, Megan J. Wallace, Nigel Curtis, Salvatore Pepe, Michael Cheung, David P. Burgner, Timothy Moss

**Affiliations:** 1Murdoch Children’s Research Institute, Royal Children’s Hospital, Parkville, VIC 3052, Australia; 2Department of Internal Medicine, Radboud University Medical Centre, 6525 Nijmegen, The Netherlands; 3School of Public Health and Preventive Medicine, Monash University, Clayton, VIC 3800, Australia; 4The Ritchie Centre, Hudson Institute of Medical Research, Clayton, VIC 3168, Australia; 5Fakultas Kedokteran, Universitas Indonesia, Jakarta 16424, Indonesia; 6Department of Obstetrics and Gynaecology, Monash University, Clayton, VIC 3800, Australia; 7Department of Paediatrics, Monash University, Clayton, VIC 3800, Australia; 8Department of Paediatrics, University of Melbourne, Parkville, VIC 3010, Australia

**Keywords:** atherosclerosis, neonatal vaccination, BCG, inflammation, *ApoE*
^−/−^

## Abstract

**Simple Summary:**

Bacille-Calmette Guérin (BCG), the vaccine against tuberculosis, is the most widely used vaccine in the world, given to almost two-thirds of newborns. BCG also has non-specific effects, which affect immune responses more broadly and impact mortality from unrelated infections. It is also important to understand the effects of BCG on other immune-related diseases, such as the development of cardiovascular disease. This has previously been studied in numerous animal studies, but not with an equivalent protocol and BCG dosage to human newborn vaccination. In this study, we vaccinated newborn mice with BCG using a dose, timing and administration route similar to human newborn vaccination. We show that BCG decreases atherosclerosis, both the number of atherosclerotic plaques as well as inflammation within the plaque. Translating our findings to humans, these potentially beneficial effects might be enhanced, as BCG vaccination decreases all infections, and infections are also associated with cardiovascular disease, so BCG could further lower the risk of developing cardiovascular diseases.

**Abstract:**

Bacille-Calmette Guérin (BCG) modulates atherosclerosis development in experimental animals, but it remains unclear whether neonatal BCG vaccination is pro- or anti-atherogenic. Many animal models differ fundamentally from BCG administration to human infants in terms of age, vaccine preparation, dosing schedule, and route of administration. We aimed to elucidate the effect of neonatal subcutaneous BCG vaccination—analogous to human BCG vaccination—on atherosclerosis development in *ApoE^−/−^* mice. At 2 days of age, a total of 40 *ApoE^−/−^* mice received either a weight-equivalent human dose of BCG, or saline, subcutaneously. From 4 weeks onwards, the mice were fed a Western-type diet containing 22% fat. At 16 weeks of age, mice were sacrificed for the assessment of atherosclerosis. Body weight, plasma lipids, atherosclerosis lesion size and collagen content were similar in both groups. Atherosclerosis lesion number was lower in mice that received BCG. Macrophage content was 20% lower in the BCG-vaccinated mice (*p* < 0.05), whereas plaque lipid content was increased by 25% (*p* < 0.01). In conclusion, neonatal BCG vaccination reduces atherosclerosis plaque number and macrophage content but increases lipid content in a murine model of atherosclerosis. Human epidemiological and mechanistic studies are warranted to investigate whether neonatal BCG vaccination is potentially atheroprotective.

## 1. Introduction

Cardiovascular disease (CVD) remains the leading cause of death worldwide [1]. Atherosclerosis, the underlying pathogenesis, is a chronic inflammatory process that develops from early life onwards [2]. There is increasing interest in understanding potentially modifiable inflammatory pathways to ameliorate the development of atherosclerosis and reduce the risk of plaque rupture [2]. In this context, Bacille-Calmette-Guérin (BCG), a vaccine designed to protect against tuberculosis, is of particular interest. BCG is the most frequently administered vaccine worldwide; approximately 85% of all infants globally are vaccinated in the early neonatal period [3]. BCG has heterologous (non-specific) immunomodulatory effects associated with reduced all-cause mortality in neonates [4], in part through the induction of a non-specific memory phenotype of the innate immune system (‘trained immunity’) [5,6]. Trained immunity may contribute to the well-described association between infections and atherosclerosis, [7] and BCG may have indirect anti-atherogenic effects by reducing childhood infection burden, as well as by modulating innate immune responses.

Studies of BCG and atherosclerosis are limited to animal models, in which BCG administration differs substantially from that in human infants in terms of age at immunization, BCG preparation [8], dose [9], schedule, and anatomical vaccination site [10]. Moreover, these experimental animal models also vary considerably in other aspects of their methodology. Studies have used: chickens [11], rabbits [12], and atherosclerosis-prone genetically modified mouse strains [13]; live or freeze-dried BCG [8]; single and repeated [12] immunizations; immunizations at varying ages following weaning; weight-equivalent doses that differ by approximately 10-fold; and intranasal [10], intraperitoneal [14], and intravenous [9] administration. The findings from studies using these diverse experimental conditions have been inconsistent, and it is difficult to extrapolate these animal data to humans, in whom BCG immunization is given once, subcutaneously with a live inoculum in the early neonatal period.

We aimed to address some of these methodological shortcomings and investigate the effect of BCG vaccination on atherosclerosis under conditions that more closely resemble human BCG vaccination: a single subcutaneous vaccination with a human dose-equivalent of live BCG inoculum to two-day-old *ApoE*^−/−^ mice, a widely used murine model, in which atherosclerosis develops in response to a high fat diet [15]. We compared atherosclerosis lesion extent and severity, lesion lipid and macrophage cell content, and plasma lipids in BCG vaccinated and control (saline-treated) adult *ApoE*^−/−^ mice (16 weeks of age).

## 2. Materials and Methods

### 2.1. Experimental Setup and Mice

Ethical approval was obtained for all animal experimentation from the relevant Monash University Animal Ethics Committee (AEC). *ApoE*^−/−^ mice on a C57Bl/6/J background were bred and held in the Monash Medical Centre (MMC) Specific Pathogen Free (SPF) animal facility until termination of the experiment at 16 weeks. At 2 days of age, subcutaneous BCG was administered as described below. At 4 weeks, pups were weaned and fed a high (22%) fat, 0.15% cholesterol ‘Western fast food diet’ (Specialty Feeds, Australia; SF00-219). At 16 weeks (mature adulthood), mice were transferred to the conventional animal facility at MMC for tissue collection (see Figure 1A for summary timeline).

Investigators administering BCG vaccine or saline injections were blinded to treatment. Immunohistochemical staining was done in single batches for each antigen to minimize variation. Each histological variable was measured by an individual investigator who was blinded to treatment.

### 2.2. Vaccine Preparation and Animal Immunization

BCG vaccine was cultured and prepared at the Centenary Institute, Sydney, Australia. Briefly, BCG was cultured in 7H9 Broth Base (Sigma-Aldrich, St. Louis, MO, USA), with 10% ADC growth supplement (Sigma-Aldrich, St. Louis, MO, USA), 0.05% glycerol, 0.05% Tween and incubated at 37 °C, 5% CO_2_. Aliquots (1 mL) of BCG in phosphate-buffered saline (PBS) were stored at −86 °C (8.47 × 10^7^ colony forming units (CFU)/mL). BCG was diluted to a final 0.5 × 10^4^ CFU/mL dose and prepared freshly on the day of vaccination. Before and after injection, pups were gently covered in sawdust from their cage to ensure their scent is familiar to mothers when returned. Two-day-old *ApoE*^−/−^ pups were given a single 10 μL injection of either BCG vaccine (0.5 × 10^4^ CFU/mL as per FDA human to mouse conversion guidelines [16]) or saline subcutaneously at the scruff of the neck, using a sterile glass syringe (Hamilton). All pups in the same litter were given the same treatment, the litters were randomized for treatment as a whole. After vaccination, the pups were monitored daily for 3 days but were otherwise left undisturbed.

### 2.3. Tissue Collection

At 16 weeks, mice were weighed, and terminal anesthesia was induced with 5% isoflurane (Cat#: FDG9623 Baxter Healthcare (Old Toongabbie, NSW, Australia)) until respiration ceased. A transverse incision was made below the costal margin. A midline incision through the sternum and abdomen was made and the diaphragm was removed. A mid-axillary incision was made lateral to the left internal thoracic artery, meeting at the lung apex to remove the rib cage. Mice were immediately exsanguinated by perforating the inferior vena cava and blood was drawn using a heparin-coated 23 G needle. Blood was centrifuged at 3000 rpm at 4 °C for 10 min. Plasma was separated and snap-frozen in liquid nitrogen for batched molecular analysis.

The heart and aorta were removed en bloc from the heart down to the iliac bifurcation. The heart and aorta were then transferred to a silicone elastomer base agar plate (Sylgard 184, Cat#: 184 SIL ELAST KIT 0.5 KG, Ellsworth Adhesives (Melbourne, VIC, Australia)) resting on ice, and were immersed in normal saline (0.9%). Using fine dissecting forceps (Dumont & FilsCie Succ #5 Inox Biologie, Agnthos, Sweden, Cat No. E120-5PO) and fine spring scissors (Vannas scissors 8.5 cm CVD, World Precision Instruments, Hilton, SA, Australia, Cat No. 501232), the heart and aorta were stripped of all surrounding fat and connective tissue under a dissecting microscope (Carl Zeiss Stemi 2000-C, Macquarie Park, Sydney, NSW, Australia).

#### Aortic Sinus Collection

The aortic sinus was collected from all mice for histological analysis by dissecting the heart, from the base of the atria and the beginning of the brachiocephalic artery (BCA) and embedded in Optimal Cutting Temperature (OCT) compound. All collected tissue was stored at −80 °C for later molecular analysis.

### 2.4. Plasma Analysis of Lipids

Plasma cholesterol and triglycerides were measured using a Cholesterol E resp Triglyceride E test according to the manufacturers’ instructions (Fujifilm Wako Pure Chemical Corporation, Heidelberg West VIC Australia, 439-17501 resp 432-40201). HDL was first isolated by adding 1/10 Dextran solution to the plasma (1:1 1 M MgCl_2_ and Dextraslip 50 (Sigma, D8787-1G) 20 g/L (2% solution)). Samples were centrifuged at maximum speed (30 min, 4 °C) and cholesterol levels of the supernatant were subsequently measured using the Cholesterol E test.

### 2.5. Histological and Immunohistochemical Assessment of Atherosclerosis in the Aortic Sinus

Aortic sinus embedded in OCT was sectioned at 5 μm using a cryostat. Four consecutive sections were taken per analysis at the aortic sinus and the mean of 4 sections were used for each analysis. Sections were mounted onto glass microscope slides (Superfrost™ Plus, Menzel-Gläser, Thermo Scientific, Scoresby, VIC, Australia) for histological and immunohistochemical staining.

#### 2.5.1. Hematoxylin and Eosin Staining

Slides were immersed in filtered Mayer’s Hematoxylin Solution (Cat#: MH-1L, Amber Scientific (Chem Supply Australia, Gillman, SA, Australia)) for 3 min and rinsed three times with distilled water. They were then counterstained with Eosin (Eosin 1%, Cat#: 0458/500ML, Amber Scientific (Chem Supply Australia, Gillman, SA, Australia)) and dehydrated with 70% ethanol and 100% ethanol twice and subsequently washed with xylene thrice. Finally, the slides were covered with coverslips using DPX and left for 24 h to dry. Slides were then scanned and lesion number and stage were assessed on each slide. Determination of lesion stage (stages I–V) was supported by Masson’s Trichrome staining (see below) to assess fibrous cap development and was performed according to the American Heart Association (AHA) guidelines [17]. Lesion size was measured by subtracting the area of aortic sinus with the lesion, from the total area of aortic sinus.

#### 2.5.2. Masson’s Trichrome Staining

Fibrous cap development was visualized by staining tissue with Masson’s Trichrome (Cat#: TRIC-500, Amber Scientific (Chem Supply Australia, Gillman, SA, Australia)). Images were obtained and analyzed using Fiji (Image J, version 2.0, open source at imagej.net, accessed on 4 September 2022) to identify collagen, which is stained blue. Nuclei of cardiac muscle cells were stained purple and subsequently cropped out in Fiji. Image analysis was divided into two parts, aortic sinus area measurement and fibrous cap development. The total lesion area was obtained by expressing the area of the image with lesion as a proportion of the total tissue area.

#### 2.5.3. Oil Red O Staining

Oil Red O was used to quantify lipid deposition in vessels, according to the manufacturer’s protocol (Sigma Aldrich St. Louis, MO, USA #MAK194). Aortic sinus cryosections (5 μm) were fixed on slides in 10% neutral buffered formalin for 10 min. Slides were then treated in the following sequence: (Oil Red O working solution stain (1 h), 60% isopropanol (2 min), water rinse and counterstained with Mayer’s Haematoxylin (3 min) and then mounted (Fluorescence Mounting Medium, Dako, North Sydney, NSW, Australia #S3022380-2) and coverslipped. Lipid (Oil Red O-stained area) area was quantified in TIFF images of stained sections using Fiji (Image J, version 2.0). Each image was scaled using a known distance. The red stain was the Region of Interest (ROI). Using the max entropy setting, red color threshold was set at a minimum of 0 and maximum of 200. The free-hand tracing tool was used to quantify the total aortic tissue area. Oil Red O-stained area was expressed as a proportion of the total lesion area.

#### 2.5.4. F4/80 Immunohistochemistry

F4/80 immunohistochemistry was used to detect plaque macrophage content. Slides containing 5-μm sections of aortic sinus were air dried for 20 min and then hydrated in water. Slides were incubated with 3% H_2_O_2_ peroxidase blocking solution (Dako, North Sydney, NSW, Australia, S2023) for 10 min to block endogenous peroxidase activity. Non-specific binding was reduced with Dako serum-free protein block (Dako, North Sydney, NSW, Australia, X0909) for 10 min at room temperature. Sections were incubated in rat α-mouse F4/80 primary monoclonal antibody (1:200, CI:A3-1, ab6640) for 1 h at room temperature (RT).

Sections were rinsed in wash buffer (Dako, North Sydney, NSW, Australia, K8024) and secondary goat α-rat adsorbed (1:500; Vector Laboratories, distributed by Abacus, Meadowbrook, QLD, Australia, BA-9401) antibody was applied to tissue for 30 min at RT. After a rinse with wash buffer, tissue was incubated with streptavidin HRP (1:500; Dako, North Sydney, NSW, Australia, P0397) for 30 min at RT, followed by DAB chromogen (Dako, North Sydney, NSW, Australia, DAB+ chromogen in substrate buffer, K5007) for 10 min. Slides were counterstained with haematoxylin for 5 min (Dako, North Sydney, NSW, Australia, S3301), washed under running tap water and subsequently placed in Scott’s tap water for 30 s. After a final rinse under running water, slides were dehydrated through a graded series of alcohol and cleared through a series of xylene and then coverslipped in Leica CV mount (Cat:14046430011) mounting medium. F4/80-stained area was expressed as a proportion of the total lesion area.

### 2.6. Statistical Analysis

A Shapiro–Wilk test of normality was performed and QQ plots were generated for each data set. The variance was assessed using the F-test. Normally distributed data of equal variance were analyzed using an unpaired Student’s *t*-test. Non-parametric data were compared using a Mann–Whitney U-test. Sex effects were assessed using two-way ANOVA. All data are presented as mean ± standard error of the mean (SEM) except for the lesion stage data, which are ordinal and thus presented as median and interquartile range. All statistical analysis was done using SPSS Statistics version 22.0 (IBM corp, 2013, Armonk, NY, USA) and graphs were generated using Graphpad Prism ^®^ (Prism 9 for Mac OS X, v9.0e, San Diego, CA, USA).

## 3. Results

### 3.1. Baseline Characteristics

A total of 7 pregnant *ApoE*^−/−^dams gave birth to 40 pups, which were randomly assigned to either the Saline injected control group (*n* = 20, 4 dams) or the BCG-treated group (*n* = 20, 3 dams, Figure 1A) on day 2. A total of 5 pups did not survive in the Saline group; one pup did not survive in the BCG group (see cause of death in Appendix A). The final Saline group contained 60% females (*n* = 9, depicted in green in all figures) and the BCG group 37% (*n* = 7). Male mice are depicted in blue in all figures. At 16 weeks of age, body weight (Figure 1B), plasma total cholesterol (Figure 1C), HDL cholesterol (Figure 1D) and plasma non-HDL cholesterol (Figure 1E) were similar between groups. Plasma triglycerides (Figure 1F) were 57% higher in BCG-treated mice at week 16 compared to control mice (*p* = 0.03). In line with previously described sex-differences in murine body weight [18], body weight was higher in male mice than females (Appendix A). Total cholesterol was higher in BCG female mice than male mice but not in saline-treated mice (Appendix A). There were also differences between groups and sexes in non-HDL-C (Appendix A), in line with sex differences described for this animal model before [19]. HDL-C and triglycerides were similar between groups and sexes (Appendix A) although numbers were too limited for some sub-groups to reach any statistical significance.

### 3.2. Neonatal BCG Vaccination Decreases Lesion Number, with Increased Lesion Lipid Content and Decreased Macrophage Content in the Plaques

Atherosclerosis lesion size (Figure 2A) and collagen content (Figure 2B) were similar between treatment groups and sexes (Appendix A).

Lesion number was lower in BCG-treated mice compared to saline-treated (*p* = 0.05, Figure 2C). Lesion stage was similar in BCG-treated mice compared to saline-treated mice (Figure 2D,E and Appendix A). In BCG-treated mice, lipid accumulation in the aortic sinus was 25% higher than in saline-treated control mice (*p* < 0.01) (Figure 2F and representative images in G); similar findings were observed in sex-stratified analyses (Appendix A). We observed some correlation between Oil-Red-O staining and plasma triglyceride concentrations, although this did not reach nominal statistical significance (*p* = 0.06, Appendix A).

F4/80 staining of macrophages in lesions of BCG-treated mice was 20% lower than in the control group (*p* < 0.05, Figure 2H and representative images in I). This difference was most evident in female mice (Appendix A, 27% decrease in female mice compared to 1.4% decrease in male mice). F4/80 staining correlated negatively with levels of non-HDL cholesterol in all mice (R = −0.41, *p* < 0.05, Appendix A) but not with triglycerides, total cholesterol or HDL cholesterol.

## 4. Discussion

Vaccination of neonatal *ApoE*^−/−^ mice with BCG (equivalent in timing and dose to administration in humans) had favorable effects on atherosclerotic lesion development at an age equivalent to early adulthood in humans, independent of plasma lipids or body weight. Lipid accumulation in the aortic sinus was increased in BCG-vaccinated mice, but macrophage content was lower, which is associated with a more stable plaque phenotype [20]. If analogous atheroprotective effects occur in humans, neonatal BCG vaccination may have beneficial but largely unappreciated effects on CVD risk [21].

Results from animal studies on the effects of BCG on atherosclerosis are inconsistent. Experimental approaches vary markedly by model species [9,11,12], live versus killed [8,10] BCG inoculum, age [10], and route of administration [9,10,14]. Some reports indicate that BCG vaccination may have proatherogenic effects [10], albeit under experimental conditions that differ substantially from BCG vaccination in humans, particularly with respect to dose, schedule, route of administration and age at immunization (See Appendix A for an overview). For example, in juvenile rabbits, two subcutaneous injections of BCG at 10 weeks (approximating to a full human BCG dose) and 14 weeks of age (half human dose), followed by a high cholesterol diet, resulted in systemic immune activation and increased aortic monocyte recruitment and atherogenesis, without differences in plasma lipids [12]. In 14-week-old female *ApoE*^−/−^ mice administered varying doses of intraperitoneal heat-killed BCG [14], mice receiving the highest dose of 1 mg heat killed BCG also developed advanced, calcified atherosclerosis [14].

In a study investigating the association between active mycobacterial infection and atherosclerotic CVD, rather than BCG vaccination per se, intra-nasal live BCG inoculation (0.3–3 × 10^6^ CFU) of 12-week-old C57BL/6J Ldlr^−/−^ mice fed a high-fat diet induced a persistent systemic BCG infection in lung and spleen [10]. Compared to uninfected controls, BCG-infected mice had more extensive aortic atherosclerosis and immune activation (non-classical monocyte proportions and increased CD4/CD8 T cell ratio) at 8- or 16-weeks post-vaccination, with no differences in plasma lipid profile between the groups [10]. These observations are analogous to both active tuberculosis disease and latent tuberculosis infection in humans, which are associated with an increased likelihood of developing subclinical obstructive coronary artery disease [22]. Preventing tuberculosis by BCG vaccination may therefore be protective against CVD events, in addition to putative atheroprotective effects of the BCG vaccine.

Animal data also suggest that BCG reduces atherosclerosis under different experimental conditions and different species. BCG administration to three-month-old chickens reduced atherosclerosis to varying extents, depending on the site of administration (subcutaneously or intraperitoneally) and the timing relative to a high cholesterol dietary intervention, changes suggested to relate to phagocyte activation [11]. Studies in atherosclerosis-prone mouse strains also indicate that, depending on experimental conditions, BCG may have atheroprotective effects. In a study of both Ldlr^−/−^ and *ApoE*^−/−^ mice receiving either 4 or 6 doses of freeze-dried BCG subcutaneously from 6 weeks of age, aortic atherosclerosis was reduced at 30 weeks of age compared to saline-treated mice [8]. The reduction in atherosclerosis correlated with immunoregulatory changes including increases in interleukin-10, expansion of Foxp3+ regulatory T cells, and inhibition of NF-kBp65 activation [8].

Our findings suggest that in *ApoE*^−/−^ mice, neonatal BCG vaccination is atheroprotective and the effects are not mediated by induction of changes in lipid profile. There is increasing interest in the non-specific beneficial effects of neonatal BCG vaccination, and it is plausible that heterologous effects of BCG on immune responses contribute to improved outcomes. There are robust epidemiological data linking early life infection with increased cardiovascular and metabolic risk and disease [23], and trained innate immunity may underpin this relationship [7]. Neonatal BCG induces innate immune training [24] and reduces infection-related mortality in human trials [5]. BCG-induced modulation of innate immune pathways is therefore a plausible contributory mechanism, although further studies would be needed to investigate this mechanism, including those in mice not raised in germ-free conditions. In addition, our study also revealed interesting sex differences, despite small group sizes. The decrease in inflammation in plaques due to BCG was driven by a decrease of inflammation observed only in female mice. Male mice in the BCG group on the other hand had significantly higher levels of non-HDLc. From human studies, it is known that there are many sex differences in the immune system and inflammatory responses, vaccine responses and lipid biology. These aspects warrant further investigation in both murine models and human studies.

Our study has a number of strengths, particularly that we emulated as closely as possible the conditions under which in humans, BCG is given to neonates and the fact that we included sex-specific analyses. Limitations include a lack of detailed immune phenotyping, the use of F4/80 only for the identification of macrophages, and the different numbers of mice per analysis due to practical constraints. Low numbers of experimental subjects affected the power of some correlation analyses, performed to explore relationships between plasma lipids and aortic lesion measurements. Co-staining for F4/80 and neutral lipids in the same section would have been informative to study the potential contribution of non-macrophage foamy cells. Investigation of pro- or anti-inflammatory macrophage markers, and assessment of circulating inflammatory cells and cytokines warrants follow-up studies, specifically to assess the short- and long-term effects of BCG treatment on immune cells in this model.

Animal models of atherosclerotic CVD are widely used but are largely characterized by hypercholesterolemic states and do not recapitulate pathogenic mechanisms and clinical outcomes in humans. As mice do not develop atherosclerosis spontaneously, genetic manipulation is necessary for atherogenesis, including genetic deletions of either LDL-R or ApoE. *ApoE*^−/−^ mice, the most widely studied murine model, have markedly increased plasma cholesterol (largely as Very Low Density Lipoproteins (VLDL) rather than as LDL as in humans) compared to wild type C57Bl/6 mice, even when on a normal or low-fat diet [15]. In this context, it is difficult to differentiate specific immunologic determinants, particularly in advanced atherosclerosis, which we studied here after 12 weeks of the Western diet. Additional, earlier time points may be informative in future studies to study early atherosclerosis development. ApoE also has anti-inflammatory and anti-oxidative properties, which may impact atherogenesis [25]. Future studies would benefit from including mice on an LDL-R^−/−^ background.

In conclusion, in a murine model of atherosclerosis, vaccination with a single subcutaneous dose of BCG soon after birth reduced the severity of atherosclerosis in 16-week-old adult *ApoE*^−/−^ mice compared to control mice. As BCG is given to the majority of the global population in infancy, human studies are needed to investigate whether BCG is atheroprotective. Randomized trials of neonatal BCG (e.g., MIS-BAIR [26]) provide an opportunity for these investigations, as do observational population-based studies in countries (such as Sweden) where universal newborn BCG vaccination was discontinued at a single time point [27].

## Figures and Tables

**Figure 1 biology-11-01511-f001:**
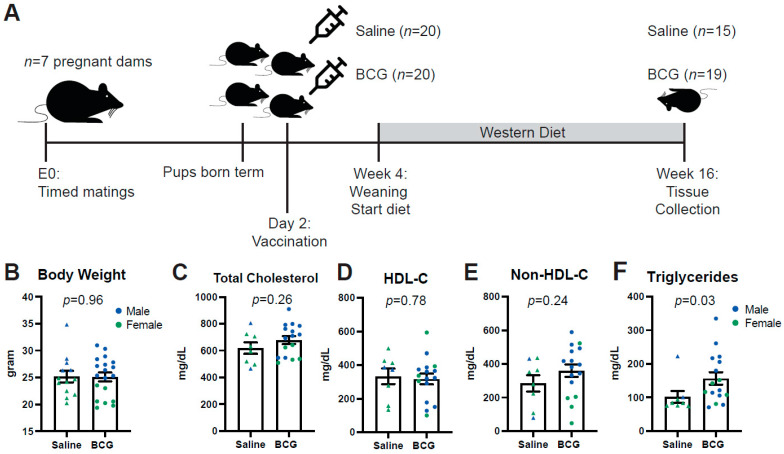
Study design and baseline characteristics. (**A**) Design of the study. 40 pups were randomly assigned to subcutaneous BCG or Saline injection at postnatal day 2. In week 4, mice were fed a Western diet for 12 weeks until tissue collection at week 16. (**B**) Body weight of the mice at week 16 did not differ between BCG or Saline treated mice; *n* = 15 for Saline (depicted as triangles) and *n* = 19 for BCG (depicted as circles). Male mice are depicted in blue, female mice in green. (**C**) Total plasma cholesterol, (**D**) HDL and (**E**) Non-HDL-C at week 16 were all similar between groups. (**F**) triglycerides were significantly higher in BCG-treated mice; *n* = 9 for Saline and *n* = 15 for BCG.

**Figure 2 biology-11-01511-f002:**
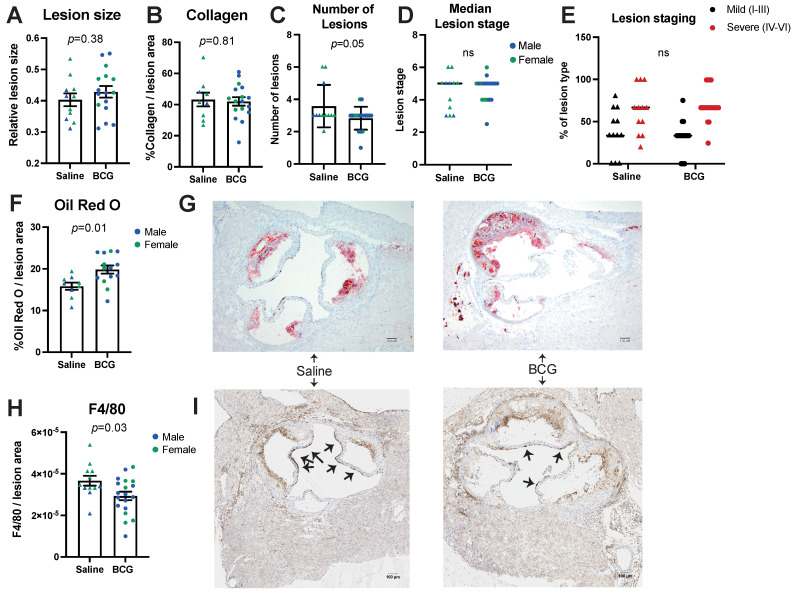
Analysis of atherosclerosis severity in aortic sinus of *ApoE*−/− mice. Atherosclerosis was assessed using several measures: (**A**) Atherosclerotic lesion size, (**B**) Collagen content and (**C**) Number of lesions as well as (**D**) median lesion stage between groups and (**E**) % type of lesion stage for each plaque (mild (black) or severe (red)); *n* = 11 for Saline (depicted as triangles) and *n* = 16 for BCG (depicted as circles). The number of lesions was significantly decreased in BCG-treated mice, whereas the median lesion stage was similar between groups. In (**A**–**D**,**F**,**H**); males are depicted in blue, females in green. (**F**) Aortic Sinus plaque lipid content was measured by Oil Red O staining and was increased in BCG-treated mice; *n* = 8 for Saline, *n* = 12 for BCG. (**G**) Representative Oil Red O images of the aortic sinus, from Saline and BCG treated mice. (**H**) F4/80 macrophage staining of plaques was significantly decreased in BCG-treated mice; *n* = 12 for Saline and *n* = 19 for BCG. (**I**) Representative images of F4/80 staining (some staining indicated by arrows).

## Data Availability

The data that support the findings of this study are available from the corresponding author upon request.

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
