# Peer review of "Neonatal Subcutaneous BCG Vaccination Decreases Atherosclerotic Plaque Number and Plaque Macrophage Content in ApoE−/− Mice"

_biology, 2022, doi:10.3390/biology11101511_

Round 1

Reviewer 1 Report

Bekkering, Singh, et al. submitted a manuscript that defines the benefit of neonatal BCG vaccination in the development of atherosclerosis. This is an important area of clinical investigation. BCG modulates atherosclerosis development in experimental animals, but whether neonatal BCG vaccination is pro- or anti-atherogenic remains unclear. The authors conducted a study of a total of 40 mice and aimed to elucidate the effect of neonatal subcutaneous BCG vaccination. They showed that neonatal BCG vaccination reduces atherosclerosis plaque number and macrophage content but increases lipid content in a murine model of atherosclerosis.  Overall, the data are important and interesting though the component linked to trained immunity and atherosclerosis is not novel. However, the conclusion needs more verification and mechanistic investigations require additional studies to prove the conclusions.

Major Comments

1. please improve the image quality of Fig2. I don’t see the significance of the Lesion number (p=0.05, Fig 2C) from the image. please show its statistics method. please use the scatter plot with each plot representing each animal instead of the violin plot.

2.  Oil Red O is a lysochrome diazo dye that stains neutral lipids and visualizes atherosclerotic plaques with an intense red color, which assists lesion quantification. Fig.2F shows that lesion area was increased in BCG-treated mice. it's the opposite effect of BCG on atherosclerotic in number. how could you explain this? would it be an experimental error from the location of the section, experimenter factor, or animal gender difference?

3.  There are 60% female animals in the saline group and 37% females in the BCG group. As we know, men are more likely to develop atherosclerosis. It’s possible that the significance of the development of atherosclerosis is due to the sex difference. Even though the authors' data kindly showed the gender separately, the data is not very solid, I’d suggest the authors add another batch to balance the number difference in gender and to validate their findings.

4. Both innate and adaptive immunity plays roles in atherosclerosis, the authors should look deeper into the different immune populations/subsets, especially after vaccination.

Minors:

All the catlog# of the regents or kits should be listed, eg. the regents in MT staining.

Author Response

Major Comments

  1. please improve the image quality of Fig2. I don’t see the significance of the Lesion number (p=0.05, Fig 2C) from the image. please show its statistics method. please use the scatter plot with each plot representing each animal instead of the violin plot.

Answer: First of all, we thank the reviewer for carefully considering our manuscript and for the comments. We apologise for the bad quality of figure 2. When we uploaded it, the quality was much better. We have now uploaded a higher quality version.

We have also changed the presentation of figure 2C-E to a scatter plot to enhance readability. The statistics methods for all analyses are described in the methods section.

  1. Oil Red O is a lysochrome diazo dye that stains neutral lipids and visualizes atherosclerotic plaques with an intense red color, which assists lesion quantification. Fig.2F shows that lesion area was increased in BCG-treated mice. it's the opposite effect of BCG on atherosclerotic in number. how could you explain this? would it be an experimental error from the location of the section, experimenter factor, or animal gender difference?

Indeed, this was a completely surprising finding to us. It appears that the number of lesions is lower, whereas there are more lipids per lesion and hence the lesions are slightly larger. On the other hand, there is less inflammation. We are not completely sure how to explain this. The Oil Red O seemed to be correlated to triglycerides, which are higher in male mice. Compared to male mice, there was decreased inflammation in female mice. It is plausible  that sex differences account for relatively larger lesions in male mice. From the literature, there are a myriad of sex differences in inflammatory responses and atherosclerosis, particularly in humans. In addition, vaccine responses differ by sex. We thank the reviewer for starting a stimulating discussion. We have added some further interpretation to the Discussion on the topic (Page 8 line 315):

In addition, our study also revealed interesting sex differences, despite sub-being of modest size. The decrease in inflammation in plaques due to BCG were driven by a decrease of inflammation observed only in female mice. Male mice in the BCG group on the other hand had significantly higher levels of non-HDLc. From human studies, it is known that there are many sex differences in the immune and inflammatory responses, vaccine responses, and lipid biology. These aspects warrant further investigation in both murine models and human studies.

  1. There are 60% female animals in the saline group and 37% females in the BCG group. As we know, men are more likely to develop atherosclerosis. It’s possible that the significance of the development of atherosclerosis is due to the sex difference. Even though the authors' data kindly showed the gender separately, the data is not very solid, I’d suggest the authors add another batch to balance the number difference in gender and to validate their findings.

We agree with the reviewer that the number of males and females are significantly different between groups, an outcome that is hard to predict from pregnant dams, which is why the group size was larger than when only 1 sex would have been studied (which is often done in murine atherosclerosis studies but then the sex differences are overlooked). Nevertheless, as we thought it potentially informative to analyse the effect of sex on the outcomes, we used ANOVA to account for the differences in sexes and outcomes between groups. In addition, we show all data with sex defined scatterplots. The separated sex analyses can be found in the supplementary data. We agree that adding another batch to balance the number is a good idea, the 10 days given for rebuttal are not sufficient to perform these studies and in addition, this might enhance confounding by batch differences between other analyses (BCG preparation, analysis of atherosclerosis) since the original study was conducted in 2019.  We would suggest that further de novo experiments should be informed by our preliminary findings.

  1. Both innate and adaptive immunity plays roles in atherosclerosis, the authors should look deeper into the different immune populations/subsets, especially after vaccination.

We agree that both innate and adaptive immunity plays a role in atherosclerosis, especially after vaccination, where the adaptive immune system is implicated, especially with vaccines other than BCG. However, our relatively modest study setup and sample collection did not allow for more in-depth analysis of the different immune populations and subsets. We have described this as a limitation in the discussion and highlighted the need for further analyses.

Minors:

All the catlog# of the regents or kits should be listed, eg. the regents in MT staining.

We apologise for the missing cat numbers. We have added them in the manuscript.

Reviewer 2 Report

The article deals with an appealing topic that provides knowledge of arteriosclerosis and possible treatments. Below I detail some aspects that I believe could improve it.

In the figure 1A there is a mouse image placed upside down.

Figure 2 is not clear, it is blurred.

I have some doubts about the graphics and the statistical analysis. I would like to know if you have analyzed the data separately for males and females and what they represent in the graphs if it is the average of males and females together. Show this in the article to clarify its understanding.

Author Response

The article deals with an appealing topic that provides knowledge of arteriosclerosis and possible treatments. Below I detail some aspects that I believe could improve it.

In the figure 1A there is a mouse image placed upside down.

We thank the reviewer for reading our manuscript and providing suggestions for improvement. We have actually meant the mouse to be upside down to show the end of the protocol. If the reviewer believes the mouse should be shown differently, we’re happy to change it back.  

Figure 2 is not clear, it is blurred.

We apologise for the blurred figure. When we uploaded the figures, it was much higher quality. We have now uploaded a better quality figure

I have some doubts about the graphics and the statistical analysis. I would like to know if you have analyzed the data separately for males and females and what they represent in the graphs if it is the average of males and females together. Show this in the article to clarify its understanding.

We have analysed the data both together as well as separately for males and females; the latter are provided in the supplementary data/figures. We thought it was of interest to show the males and females together in the overall figures, to highlight sex differences. We refer to these supplementary figures and analyses throughout the manuscript.

Reviewer 3 Report

This study focuses on evaluating the effects of neonatal subcutaneous BCG vaccination on atherosclerosis development n ApoE-/- mice. Soon after birth, the mice received either a weight-eqiuvalent human dose of BCG or saline control and later were fed a Western-type diet containing 22% fat, and were sacrificed for assessment at 16 weeks of age. The manuscript shows that some features, such as body weight, plasma lipids, atherosclerosis lesion size, and collagen content, were not affected by the BCG vaccination, while some features were sigficantly affected. Atherosclerosis lesion number and macrophage content both decreased in the BCG-vaccinated mice, while plaque lipid content increased. 

Before this study, there were studies exploring the effects of BCG on atherosclerosis, using different experimental settings or methodologies, such as different animal models, BCG preparation, doses, and anatomical vaccination sites, and the conclusions have been inconsistent. This study used the BCG vaccination condition that is more similar to human BCG vaccination, and contributes to a better understanding of whether or/and how BCG vaccination affects atherosclerossis. 

Overall, this manuscript is well prepared and data and conclusions are well stated. The parts that need attention or could be improved are as following. 

(1) Some data in Suppl Fig 1 are confusing. In panel A, there are five data points in the group of Saline-M. However, in panels B-E, the group of Saline-M only has 2 data points. This is both confusing and needs to be treated more cautiously, especially considering the fact that in panels B, C, and E, the variations between the 2 data points are significant.

(2) The manuscript shows and mentions the gender difference in terms of the body weights and tryglyceride levels. However, no discussion was provided. It would be better if any explanations or discussions could be added to this point.

Author Response

(1) Some data in Suppl Fig 1 are confusing. In panel A, there are five data points in the group of Saline-M. However, in panels B-E, the group of Saline-M only has 2 data points. This is both confusing and needs to be treated more cautiously, especially considering the fact that in panels B, C, and E, the variations between the 2 data points are significant.

The reviewer is correct that there are some data points missing for the cholesterol measurements due to technical difficulties. Unfortunately, not all plasma samples had sufficient volume to assess the cholesterol levels. We agree that this raises the possibility that the true mean for this group is hard to estimate, as the values are highly variable. In the discussion – limitation section (page 8 line 327) we have mentioned this limitation. We have added another note of caution on page 5 line 217-218 (in the results section). For all the other groups and measurements, there were sufficient data points to assess group differences. 

(2) The manuscript shows and mentions the gender difference in terms of the body weights and tryglyceride levels. However, no discussion was provided. It would be better if any explanations or discussions could be added to this point.

The reviewer is correct that those significant differences were not discussed in the current manuscript. For both body weight and triglycerides in mice, it is known that there are sex differences: male mice are heavier than female mice and females have lower triglycerides. We have now added references to show that our findings are in line with previous findings in this specific mouse model.

Round 2

Reviewer 1 Report

Thanks to the authors' responses. I don't have further questions.